# Experiences and Lessons Learned from Implementing the RELIEF Digital Symptom Self-Reporting App in a Palliative Home Care Setting

Jehanara Chagani [1], Donny Li [2,3,4,*], Bonnie Keating [5] and Martin Chasen [2,6,7]

1   Central West Local Health Integration Network, Brampton, ON L6W 4P3, Canada
2   Department of Palliative and Supportive Care, William Osler Health System, Brampton, ON L6R 3J7, Canada
3   Faculty of Health Sciences, McMaster University, Hamilton, ON L8S 4L7, Canada
4   Department of Research, Humber River Hospital, Toronto, ON M3M 0B2, Canada
5   19 Blacksmith Road, Oak Bluff, MB R4G 0A2, Canada
6   Department of Family Medicine, McMaster University, Hamilton, ON L8S 4L7, Canada
7   Department of Family and Community Medicine, University of Toronto, Toronto, ON M5S 1A4, Canada
*   Correspondence: donny.li@williamoslerhs.ca

**Abstract:** The majority of Canadians agree they have the right to end-of-life care in their own homes. While a palliative approach to care in the home setting has been demonstrated to be beneficial for patients and the healthcare system, it has rarely been well-integrated through an eHealth approach. Thus, in 2018, we piloted the RELIEF app, a digital symptom self-reporting tool for patients with palliative care needs. This was followed by the initiation of an extension phase of RELIEF in the home care setting. In this commentary, we share the implementation perspectives and experiences of the researchers and healthcare workers involved in this home care phase. It was mainly expressed that there were challenges with nurses feeling involved, supporting the research program, and using the technology, while patients and family caregivers had challenges using the app and cooperating with staff. We describe our lessons learned from these experiences and future changes to be enacted. A detailed report of this trial will be made available in future publications.

**Keywords:** palliative care; home care; symptom reporting; remote monitoring; virtual care; community health; eHealth

## 1. Introduction

Over 90% of Canadians agree patients have the right to receive care in their own homes at the end of life—greater than half of Canadians anticipate the majority of their end-of-life care to take place in their homes [1]. Indeed, there are numerous demonstrated benefits to a palliative approach to care in a patient's own home [2]. Receiving palliative care services at home has been associated with a 47% reduction in palliative patient deaths in the hospital, thus also avoiding additional healthcare costs. A major challenge in achieving these outcomes has been the insufficient resources and services for timely patient symptom assessment, monitoring, and management. Poor symptom monitoring in the home and late referrals for symptom management may account for many hospital decedents [3].

Previous studies have demonstrated that remote monitoring and automated collection of patient self-reported symptoms can improve care for people with palliative care needs [4–6]. However, existing digital solutions are rarely well-integrated into clinical workflow [7]. Thus, we co-created the digital RELIEF app (version 3.7, uCarenet Technologies Inc. Toronto, ON, Canada): a web-based application for symptom self-reporting by patients with palliative care needs. The patient or caregiver securely logs into the site daily to self-report symptoms, distress, and pain using a set of three validated clinical symptom assessment measures: (A) the Edmonton Symptom Assessment System Revised (ESAS-r); (B) the Distress Thermometer (DT); (C) the Brief Pain Inventory (BPI). The data

is available in real time to the patient's circle of care via RELIEF. Increases in symptom burden, distress, or pain, as quantified by the numerical rating the patient selects in the ESAS-r and DT assessments, are detected and analyzed by an evidence-based computer algorithm designed and approved by clinical experts. The algorithm will generate an alert when a patient reports: (1) an increase of 2 points each day over 2 consecutive entries; (2) an increase of 3 points over the previous entry; or (3) any score of 8 or higher, for any of the symptoms listed in the ESAS-r and DT. Following review of the RELIEF alert by the healthcare provider(s), patients receive: (1) earlier intervention; (2) mobilization of auxiliary services; and as needed (3) recommendation for urgent clinic/home visit or the emergency department. The app is currently available in both English and French, but it is anticipated that more languages will be offered in the future.

In a previously published RELIEF pilot study, we demonstrated the feasibility of the RELIEF app as a tool for patients with palliative care needs to self-report their symptoms, receive timely interventions, and minimize unnecessary visits to the emergency department (ED) or admissions to the hospital [8]. A second pilot study focused on the use of RELIEF in the community/home care setting and was conducted with the patients in their home with home care nurses. These nurses were employed by various nursing agencies, visiting their patients on a schedule. Patients were to call the nursing agency if there was any emergency or need for immediate care—nurses then connected with the physician if required. For the purpose of research, the team selected a geographic area and a specific nursing agency with the greatest number of patients with palliative care needs. Initially, five primary palliative care nurses from this agency were trained on how to use the app, and each was asked to identify five eligible patients who the research coordinator may contact. Similar to the previous study, when patients completed RELIEF assessment forms (ESAS-r, DT, BPI) that indicated a need for comprehensive symptom assessment, support, and management, an alert was generated and received by that patient's nurse, who was expected to respond to them within one hour of receiving the alert. In this commentary, we describe the challenges and lessons learned from the implementation of the community RELIEF pilot trial thus far.

## 2. Nursing and Staffing Challenges

### 2.1. Nurse Involvement, Communication, and Workload

The patients enrolled had low symptom burdens, thus requiring minimal support from their circle of care and infrequent nursing visits. A detailed discussion with the nurses identified that some nurses were not aware of their participation in the study prior to training, nor were they included during the planning phase.

Nurses' patient loads were increased up to double due to staff shortages as a result of the COVID-19 pandemic. This was further compounded by the specific turnover of staff who were involved with implementing the RELIEF study. In addition, the time that remaining nurses were required to spend with each patient also increased due to pandemic protocols.

Some nurses were under the impression that the alerts were a measure of their performance and would negatively impact them if they could not respond to patients within a one-hour period. They felt that receiving the alert and responding to it promptly with their busy schedules and while caring for other patients would not be possible. This perception was problematic as it led to the recruitment of patients with low symptom burden and could have contributed to the slow recruitment process.

### 2.2. Nurse Support for RELIEF Research

Some nurses perceived that their work was not valued as they did not feel supported by their team, mainly by their physician colleagues and management, when they faced challenges in the community and with their patients. One nurse shared an example that when she was struggling to simultaneously provide care to patients and sought the help of the physician on call, she felt especially discouraged when the physician responded that it was not a problem of their concern. As a result, many nurses were hesitant to engage in

additional work, such as participation in research projects beyond their community nurse role/job description.

Another factor that may have played a role in nurses' low participation in the research is that since nurses mostly enrolled patients with low symptom burdens, they were less able to see the impact the app had on the timely provision of care. One of the nurses who had a patient with a higher symptom burden reported observing a substantial difference in the quality of care provided since the RELIEF app allowed the patient to receive timely care in the final days of life, allowing them to die peacefully at home. This difference was also observed by the family caregiver, who felt reassured and comforted.

When the five original nurses required additional help with their research duties, the agency agreed to enlist the participation of all nurses working in that particular geographic area in order to facilitate patient enrollment. Through this experience, it was identified that these additional nurses had also found it difficult to realize the potential impact of the study, specifically the use of the app in providing timely care to patients.

*2.3. Technology*

Since the nurses did not have work phones and were only provided laptops by their organizations, they had challenges switching between their work and personal emails to check RELIEF notifications, often requiring them to log in and out of different email platforms multiple times per shift. There were many instances when nurses did not feel comfortable using their personal phones for work-related purposes. Although using their work laptops may resolve some of these issues, it was not feasible to check notifications and emails through laptops when on the road or with patients.

Even when nurses were notified of an alert, patients were only identified by their randomly generated study ID number assigned at the time of enrollment for privacy reasons. The nurses would then have to connect to the patient's electronic medical record (EMR) to review the patient's chart and support their symptom assessment and management. This process, though inefficient, was the only course of action possible as the participating organizations did not have the resources to assign a central person to monitor the app and alerts.

Additional technological challenges include emails that were blocked due to organization firewall safety settings—this made receiving alerts unreliable for nurses. In all, these technological barriers remained even after involving management staff, which could partially have contributed to the challenge of nurse support throughout the study period.

*2.4. Lessons Learned and Future Recommendations*

The challenges faced by the team allowed for many lessons to be learned. One lesson was to engage nurses from the inception of the project and involve them in the development of the protocol. This would be important in maximizing their participation and planning accordingly for potential barriers identified from their lived experiences of nursing in the community and the privacy of patients' homes. Engaging at least one nurse and having a champion in the organization who could advocate for the functionality and benefits of the app while correcting misinformation could have improved nurse support for RELIEF research, and consequently, positively impacted the enrollment of patients. The research team also felt that a site checklist outlining the material and contractual requirements for the study, developed prior to initiating the project, would have been beneficial as it may prevent issues with expectations, such as nurses not having the technological requirements for the study. Having a formal contract in place between agencies may also be beneficial since it would have firmed the commitment of the service provider organization to the project such that staff/manager turnover would not have significantly impacted the integrity and continuation of the study. It was also identified that the initial study training provided to nurses should emphasize that participation in this study and its outcomes will not affect their employee performance evaluation—this point should be consistently reinforced by

the champion. Finally, periodical check-ins with nursing staff involved in RELIEF would have helped identify critical issues early on while further engaging nurses.

## 3. Patient and Family Caregiver Challenges

### 3.1. Completing RELIEF Assessments

Issues arose when family caregivers were responsible to help complete the patient RELIEF assessment forms due to patients being too sick, fatigued, or not knowing how to use the device/app. These caregivers often felt that they already had a significant responsibility and would be further overburdened by this additional role. Some caregivers also reported that they found the forms too long, repetitive, or not helpful to complete every day or every other day. Further, caregivers reported it being especially burdensome to ask the many questions from the three assessment forms if the patient's disease was in an advanced state.

Some caregivers added their own perception of the patient's symptoms instead of asking the patient; for example, one of the caregivers believed that the patient was depressed and indicated this on the RELIEF app. Therefore, caregivers using their belief of the patient's condition to enter symptoms instead of consulting the patient themselves compromised the integrity of the patient's reported outcomes.

Family caregivers reported feeling guilty for 'wasting' nurses' time when nurses called to check if the patient's RELIEF symptom score increased, even when the symptom was expected or was already managed and resolved. Further, caregivers did not find the app useful when the patient was stable and had no need for frequent nursing interventions. This was cited as a frequent reason for not completing the RELIEF assessment forms as scheduled. However, caregivers commented that it was very helpful when they had a nurse directly contact them, as otherwise, they would be on the phone with the nursing agency for several hours to speak with the nurse.

### 3.2. Cooperation between Staff and Families

It was observed that the patients and family caregivers found it difficult to trust the research coordinator and consequently sign the study consent form when visited alone, as compared to when a joint visit was arranged with their visiting nurses. Many aged immigrant patients of certain cultures felt generally uncomfortable when they were asked to sign the consent form, as signing to them indicated legal repercussions if they could not complete the full duration of the study.

### 3.3. Lessons Learned and Future Recommendations

Family caregivers and nurses both felt that it would have been helpful if RELIEF was an app that could be downloaded on their personal device, rather than being a web-based app that they had to log-in to each time. Moreover, caregivers would have preferred to receive a notification when assessment forms need to be completed, and nurses would have preferred to receive patient alerts on a platform other than email, as well as notifications when a patient does not complete their assessment forms as scheduled. The research team also felt that it would have been helpful if a newsletter was implemented to keep partners up to date and engaged with the progress of the study. The shortage of staff that was experienced by many organizations due to staff sick leaves and turnover resulted in nurses being overworked and burnt out, which also played a role in the delayed patient enrollment and low participation of nurses. Therefore, the research team felt that the nurses supporting this research deserve much acknowledgment. Recognition of nurses' contributions, their involvement in the steering committee, and their engagement at every step of protocol development would lessen the degree of challenges faced in the study, and further empower all who are a part of this journey in making a quality palliative approach to care accessible to all.

## 4. Future Directions

Throughout the study's experience thus far, we have encountered numerous challenges, but many more lessons learned. We have also taken action on some of these lessons; for example, we have amended the study protocol such that patient participants could be identified and consented by Care Coordinators before they were enrolled. Further, a central body of nurses has been engaged to monitor the alerts received—they may assess patients and engage home care nurses if required. Although we did not meet the protocol recruitment timeframe of 25 patients within 4–6 months, after implementing these changes in the protocol, we were able to improve our patient enrollment rate and complete our recruitment target after one and a half years. A detailed report of this trial will be shared in future publications.

## 5. Conclusions

In conclusion, the RELIEF app aims to improve the accessibility of home care at the end of life by addressing insufficient resources for timely and reliable patient home monitoring. After the recruitment stage of a trial on the RELIEF app in the community setting, we report our lessons learned and recommendations for addressing nursing and staff challenges in their study involvement and workload, support for RELIEF research, and use of technology. Further, we describe the lessons and recommendations from the patient and family caregiver perspectives, namely their experiences completing the RELIEF symptom assessments and collaborating with staff. In the future, we foresee this technology as being routinely used as point-of-care for patients with palliative care needs. The current challenges are integrating these tools effectively into the care process and increasing the capacity to monitor more patients at home while empowering patients and caregivers to speak with healthcare professionals on what the patient needs. As the use of technology becomes more prevalent in healthcare, we hope that our experiences inspire and assist the implementation of other eHealth initiatives.

**Author Contributions:** J.C. and D.L. are acknowledged to have contributed equally as co-first authors. Conceptualization, J.C., B.K. and M.C.; methodology of commentary, D.L., J.C. and B.K.; writing—original draft preparation, D.L., J.C. and B.K.; writing—review and editing, D.L., J.C., B.K. and M.C.; supervision, M.C.; project administration, J.C. and M.C. All authors have read and agreed to the published version of the manuscript.

**Funding:** The original research this commentary is based upon has received funding from the Centre for Aging and Brain Health Innovation (CABHI).

**Acknowledgments:** The authors would like to thank Margo Morrison, Thalicia James, and the staff and patients from the Central West Local Health Integration Network for their crucial contributions to implementing the RELIEF project in the home care setting.

**Conflicts of Interest:** The authors declare no conflict of interest.

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
