# Peer review of "Experiences and Lessons Learned from Implementing the RELIEF Digital Symptom Self-Reporting App in a Palliative Home Care Setting"

_curroncol, doi:10.3390/curroncol29120738_

Round 1

Reviewer 1 Report

Well written manuscript.

A brief description of the patient population involved in the study is perhaps not out of place. Details could include age, gender, marital status, sources of income, disease that resulted in end of life care, the most common symptom promting a home visit, narcotic usage and the languages the RELIEF app is available in. 

Similarly, a brief description of the care giver circle is also desirable: how many physicians/doctors, nurses, family members, spouses participated in the evaluation would be welcome.

A brief descritption of the algorithm used by the app and the escalation of urgency to the care giver team might also be useful.

It is important to note if the non-medical care givers were family, friends, relatives, neighbors, etc. 

It would also be important to note if the care giver lived with the patient or lived separately. How many of them were working during the duration of observation.

'Buy in' as a phrase may be American, but may not be understood by the rest of the English speaking word. A commoner other word (Support) can perhaps be used to indicate what you mean. 

Author Response

Dear Reviewer, 

We sincerely appreciate the time you have taken to review our article. We hereby address your comments:

A brief description of the patient population involved in the study is perhaps not out of place. Details could include age, gender, marital status, sources of income, disease that resulted in end of life care, the most common symptom promoting a home visit, narcotic usage and the languages the RELIEF app is available in. 

  • Thank you for raising this important point. Unfortunately, our research on RELIEF in the community setting has only recently completed its recruitment stage, and we do not currently have that data. However, our pilot trial, where such information is available, has been referenced in this article. We have added a brief description of the languages that RELIEF is offered in, as per your suggestion.

Similarly, a brief description of the care giver circle is also desirable: how many physicians/doctors, nurses, family members, spouses participated in the evaluation would be welcome.

  • As per our previous response, the information relating to a patient and their family's demographics and the healthcare team involved is not yet available as we have only recently completed the recruitment stage. However, we reported the total number of patients who consented and were recruited into the study, as well as the timeframe under section 4: Future Directions.

A brief descritption of the algorithm used by the app and the escalation of urgency to the care giver team might also be useful.

  • We have edited our article to now specify how increases in symptom burden, distress, and pain are objectively received by the app. We have also clarified how a triggered alert escalates the healthcare team via one of three actions indicated in the article.

It is important to note if the non-medical care givers were family, friends, relatives, neighbors, etc. 

  • Thank you for pointing this out to us. We have now clarified in the paper that the non-medical caregivers were only family members.

It would also be important to note if the care giver lived with the patient or lived separately. How many of them were working during the duration of observation.

  • This is certainly an important point for consideration—however, we unfortunately do not have that data at this time. We will keep this point in mind when we write our completed findings at the conclusion of this study.

'Buy in' as a phrase may be American, but may not be understood by the rest of the English speaking word. A commoner other word (Support) can perhaps be used to indicate what you mean. 

  • Thank you for raising this point. We have accepted your advice and replaced the term "buy-in" with the more universally understood word, "support."

Reviewer 2 Report

Li et al in manuscript "Experiences and Lessons Learned from Implementing the RELIEF Digital Symptom Self-Reporting App in a Palliative Home Care Setting" present first experiences related to the implementation of the eHealth application.

This is a very interesting topic, because taking into account technological progress, it is only a matter of time before such electronic patient assistants will be in everyday and common use.

Manuscript is written correctly and well structured.

Comment:

Line 205 - Chapter 5 - Conclusions - is a typed schematic sentence from template . However, a summary paragraph should be added.

Author Response

Line 205 - Chapter 5 - Conclusions - is a typed schematic sentence from template . However, a summary paragraph should be added.

  • We thank you for pointing this out. The corrections have been made and a conclusion has been added. 

The authors of this paper sincerely thank you for your time in reviewing our article. As the digital healthcare field continues to grow, we certainly agree that technologies such as these may play a significant role in home care.

Round 2

Reviewer 2 Report

No further comments.

After changes manuscript is structured properly.